# One-Day Molecular Detection of *Salmonella* and *Campylobacter* in Chicken Meat: A Pilot Study

**DOI:** 10.3390/foods10051132

**Published:** 2021-05-19

**Authors:** Andrea Zendrini, Valentina Carta, Virginia Filipello, Laura Ragni, Elena Cosciani-Cunico, Sara Arnaboldi, Barbara Bertasi, Niccolò Franceschi, Paolo Ajmone-Marsan, Dario De Medici, Marina Nadia Losio

**Affiliations:** 1Department of Animal Science, Food and Nutrition—DIANA, Università Cattolica del Sacro Cuore, Via E. Parmense, 84, 29122 Piacenza, Italy; a.zendrini@unibs.it (A.Z.); niccolo.franceschi@unicatt.it (N.F.); paolo.ajmone@unicatt.it (P.A.-M.); 2Department of Molecular and Translational Medicine, University of Brescia, Viale Europa, 11, 25123 Brescia, Italy; 3Department of Food Safety, Istituto Zooprofilattico della Lombardia e dell’Emilia Romagna, Via A. Bianchi, 9, 25124 Brescia, Italy; valentina.carta@izsler.it (V.C.); laura.ragni@izsler.it (L.R.); elena.coscianicunico@izsler.it (E.C.-C.); sara.arnaboldi@izsler.it (S.A.); barbara.bertasi@izsler.it (B.B.); marinanadia.losio@izsler.it (M.N.L.); 4National Reference Centre for Emerging Risks in Food Safety—CRESA, Istituto Zooprofilattico della Lombardia e dell’Emilia Romagna, Via G. Celoria, 12, 20133 Milan, Italy; 5Department of Food Safety and Veterinary Public Health, Istituto Superiore di Sanità, Viale Regina Elena, 299, 00161 Rome, Italy; dario.demedici@iss.it

**Keywords:** LAMP, *Campylobacter*, *Salmonella*, poultry, foodborne diseases

## Abstract

*Salmonella* and *Campylobacter* ssp. are bacterial pathogens responsible for most foodborne infections in EU countries. Poultry serves as a reservoir for these pathogens, and its important role in the meat industry makes it essential to develop a rapid detection assay able to provide results in one day. Indeed, the rapid identification of foodborne pathogens is an important instrument for the monitoring and prevention of epidemic outbreaks. To date, *Salmonella* and *Campylobacter* screening is mainly conducted through molecular methods (PCR or real-time PCR) performed after 18–24 h long enrichments. In this study, we evaluated short enrichments (0, 2, 4, and 6 h) combined with a colorimetric loop-mediated isothermal AMPlification (LAMP) or real-time PCR to detect *Salmonella* and *Campylobacter* in poultry meat contaminated at different concentration levels (10^1^, 10^3^, and 10^5^ CFU/g). Our results show that real-time PCR allows the detection of *Salmonella* and *Campylobacter*, even after shorter enrichment times than prescribed by ISO references; particularly, it detected *Salmonella* down to 10^1^ CFU/g since T0 and *Campylobacter* from 10^3^ CFU/g since T0. Detection with LAMP was comparable to real-time PCR without the requirement of a thermal cycler and with shorter execution times. These characteristics make colorimetric LAMP a valid alternative when one-day results are needed, improving the timely identification of positive meat batches, even in the absence of specialized instrumentation.

## 1. Introduction

*Salmonella enterica* and *Campylobacter* spp. are bacterial pathogens that cause the majority of foodborne infections in EU countries, being the second and first cause of foodborne diseases in 2019, respectively. More specifically, *Salmonella* was responsible for 87,923 human cases, and *Campylobacter* spp. was reported in 220,682 clinical cases [1].

*Salmonella* is a Gram-negative, rod-shaped bacterium with a facultative anaerobic metabolism belonging to the *Enterobacteriaceae* family. The genus *Salmonella* belongs to two broad species, namely *Salmonella enterica* and *Salmonella bongori*. *Salmonella enterica* represents the most pathogenic species, and includes over 2600 serovars [2,3]. Some serovars are species-specific, while others are highly adapted to a wide range of animal hosts, and are responsible for foodborne infections, causing mild to severe enteric diseases in humans [3]. The serovars more frequently involved in foodborne transmission are *S. enterica* ser. Enteritidis (*S.* Enteritidis) and *S*. Typhimurium (including its monophasic variant), which are frequently found in food of animal origin together with other serovars, like *S.* Infantis and *S*. Dublin [1,2,3].

In Europe, most salmonellosis outbreaks and infections are linked to the consumption of poultry meat, eggs, and their derived products [2]. Indeed, some strains of *Salmonella*, including *S.* Enteritidis and *S*. Typhimurium, frequently colonize the enteric tract of avian species without detectable symptoms and can survive along the processing chain, causing contamination [4]. *Campylobacter* spp. are Gram-negative, non-spore-forming bacteria belonging to the *Campylobacteriaceae* family that includes 22 different species, among which *Campylobacter jejuni* and *Campylobacter coli* represent the main cause for human gastroenteritis. Clinical manifestations of campylobacteriosis can be severe, with possible sequelae like Guillain-Barrè syndrome, reactive arthritis, and irritable bowel syndrome [5]. The main reservoir for foodborne transmission of *C. coli* and *C. jejuni* are poultry species, and handling and eating raw or undercooked chicken meat is the main risk factor for human infection, accounting for up to 24.2% of all of the *Campylobacter* spp. infections in the EU [6].

The testing and rapid detection of pathogenic microorganisms in foodstuffs, like *Salmonella* and *Campylobacter*, is crucial to identify contaminated foods and contain the spread of the pathogen before it leads to a serious outbreak.

The PCR amplification of specific DNA and RNA fragments has become the preferred method for the detection of microorganisms in foods. In particular, real-time PCR (RT-PCR) has gained ever-growing importance in the molecular screening of foodborne pathogens.

In recent years, many other amplification techniques have been developed to improve PCR in terms of sensitivity, affordability, and rapidity [7]. Loop-mediated isothermal amplification (LAMP) is one of the most widely used isothermal methods, detecting bacteria, DNA viruses, and, recently, parasites [8,9,10,11,12]. Considering that LAMP is a simple operating assay with the ability to rapidly detect pathogens in at-risk inhibition matrices, it has great potential in the foodborne diseases field [13]. Indeed, it was successfully applied to some main foodborne pathogens, such as *Escherichia coli*, *Salmonella*, *Listeria monocytogenes*, and *Staphylococcus* spp., demonstrating the possible application in the clinical diagnosis and surveillance of infection diseases [14,15,16,17].

To be performed, LAMP requires 2/3 pairs of primers that amplify the target region through elongation via a hairpin structure with stem loops at each end [18]. A DNA polymerase featuring high strand displacement activity (e.g., the *Bst* polymerase from *Bacillus stearothermophilus*) is also required. Despite LAMP being more complex than PCR in terms of setting up and optimization, it provides several advantages: (i) thanks to the strand displacement activity of the polymerase, LAMP is an isothermal reaction, thus eliminating the need for a thermal cycler; (ii) amplification is higher and faster than PCR, occurring in less than 60 min; (iii) LAMP is less sensitive to inhibitors that normally hamper PCR (e.g., detergents, salts, or lipids); and (iv) DNA amplification can be observed at a glance, without any specific equipment, if proper chemicals are added in the reaction mix (e.g., colorimetric reporters). These benefits make the LAMP reaction a tool in all applications needing rapid tests, such as on-field analyses (e.g., on-site production quality monitoring and control of third-country imports). Considering the fact that poultry consumption reached 120 million tons in the 2016–2018 period and the role of chicken as the major reservoir for *Campylobacter* and *Salmonella* infection [1], it is pivotal to ensure the fast screening of these bacteria in poultry products to grant food safety. To date, *Salmonella* and *Campylobacter* screening by RT-PCR is completed within 24–48 h from sample receipt and registration. However, a relevant lapse of this time is used for pathogen growth (18 h for *Salmonella* and 40 h for *Campylobacter*) in a proper medium (the so-called enrichment phase). Indeed, the possibility to shorten the enrichment phase would represent an enormous time-saving option. Therefore, the objective of this study was to test the effect of short enrichment times (0, 2, 4, and 6 h) on the detection of *Salmonella* and *Campylobacter* in experimentally contaminated poultry meat and to develop a one-day workflow to improve the timely diagnosis of these two pathogens. Pathogen detection was carried out both with RT-PCR and colorimetric LAMP, the latter representing an additional rapid, cost-effective, user friendly, and well-established alternative for food and feed screening of pathogens at the point of care (POC) [8,19,20].

## 2. Materials and Methods

### 2.1. Experiment Design and Experimental Contamination

To evaluate the ability of the LAMP assay to detect *Salmonella* and *Campylobacter* spp. in poultry meat, an experimental trial was set up. Four batches of ground chicken meat, 1200 g each, were spiked with different concentrations of either *Salmonella* or *Campylobacter* spp. (two batches each). As previously described [21], for each pathogen, a bacterial suspension of three different strains was prepared (Table 1). Each strain was incubated separately in BHI broth at 37 °C for 22 ± 2 h for *Salmonella*, and in Bolton broth at 41.5 °C with 5% CO_2_ for 24 ± 2 h for *Campylobacter* spp. Then, the broths were re-incubated in fresh media at the same conditions to ensure that most of the cells were in the same physiological state. The broth concentration was titrated on blood agar, and 10^3^ CFU/mL, 10^5^ CFU/mL, and 10^7^ CFU/mL dilutions were prepared. Each 1200 g meat batch was then divided into four 300 g aliquots, three of which were contaminated with the different suspensions to obtain final contaminations of 10^1^ CFU/g, 10^3^ CFU/g, and 10^5^ CFU/g, as well as one with sterile saline solution as the negative control. From each aliquot, 25 g sampling units were prepared in triplicate and homogenized in a stomacher for 1 min with 225 mL of buffered peptone water (BPW) for *Salmonella*-contaminated samples or 225 mL of Preston broth for *Campylobacter*-contaminated samples. Bacterial suspensions cultured in the broth were used as positive controls and to evaluate the effect of matrix inhibition. All samples were then incubated as specified above. Ten mL of enrichment broth were collected in 15 mL tubes at time 0 and after 2, 4, and 6 h, and stored at −20 °C until use.

### 2.2. Pathogen Plate Count

One mL from each contaminated sample, both the positive control (bacterial suspension) and negative control, was diluted in 9 mL of BPW, and 1:10 serial dilutions were prepared. For *Salmonella* spp. plate count, 0.1 mL of each dilution was plated on a Hektoen agar solid medium and incubated at 37 °C ± 1 °C for 21 ± 3 h. For *Campylobacter* spp. plate count, 0.1 mL of each dilution was plated on a modified charcoal cefaprazone deoxychocolate (MCDD) solid medium and incubated at 41.5 °C ± 0.5 °C for 44 ± 3 h under microaerophilia conditions. The bacterial concentration of each sample was estimated by counting colonies grown in plates in which a countable number of colonies (10 to 150) was observed. Each sample was measured in triplicate.

### 2.3. DNA Extraction

The 10 mL of enrichment broth previously collected were thawed at room temperature and centrifuged at 2000× *g* for 15 min to separate residual matrices. The supernatant was collected and centrifuged at 10,000× *g* 10 min. The resulting pellet was then resuspended in 100 µL of Chelex 100 (Biorad, Hercules, CA, USA) and transferred to a 1.5 mL tube. The suspension was incubated at 56 °C for 20 min and then at 99 °C for 15 min. The tubes were then centrifuged at 12,000× *g* 5 min, and the supernatant was collected and stored at −20 °C until use.

### 2.4. Real-Time PCR

To evaluate LAMP performance, *Salmonella* and *Campylobacter* were also assayed by RT-PCR using commercial kits. Briefly, 5 μL of probe and 5 μL of extracted DNA were added to 40 μL of a Mix IQ Check kit for *Salmonella* and an IQ Check kit for *Campylobacter* (Biorad, Hercules, CA, USA). The reaction was carried out in a CFX96 thermal cycler (Biorad, Hercules, CA, USA) with the following protocol: 95 °C for 10 min, followed by 50 cycles at 95 °C for 15 s, 58 °C for 20 s, and 72 °C for 30 s (total amplification time: 90 min).

### 2.5. Colorimetric LAMP

Colorimetric LAMP assays were carried out using primers previously described by Zhuang et al. [22] for *Salmonella* and by Romero et al. [23,24] for *Campylobacter* spp. (Table 2).

The reaction mixture was set up with 12.5 μL of WarmStart Colorimetric LAMP 2X Master Mix (New England Biolabs, Ipswich, MA, USA), 2.5 μL of 10X Primer mix (Table 2), and 2 μL of DNA (PCR-grade water for negative controls) from *Salmonella* or *Campylobacter*. PCR-grade water was added to reach a final volume of 25 µL.

Colorimetric LAMP results were visualized by the naked eye, as the color of the mix shifted from bright red to yellow in the case of positive amplification. LAMP reactions were carried out at 65 °C for 30 min (hereafter referred as 30′ LAMP) and 45 min (hereafter referred as 45′ LAMP) on a heated plate equipped with 0.2 mL tube adaptors. LAMP assay specificity was also tested against some of the most common foodborne pathogens, namely *Yersinia enterocolitica, Listeria monocytogenes*, and verocytotoxin-producing *E. coli*.

### 2.6. Statistical Analysis

The concordance of LAMP and RT-PCR results was assessed by McNemar’s and binomial tests. Linear regression was used to evaluate the linearity of RT-PCR results in relation to the different enrichment times and contamination levels. Data analysis was conducted by open-source software R (3.4.3 version).

## 3. Results

### 3.1. Pathogen Plate Count

According to pathogen plate count, both *Salmonella* and *Campylobacter* grew as expected. In *Salmonella*-contaminated samples, the pathogen showed exponential growth beginning after 2 h of enrichment (Figure 1a,b). Indeed, *Salmonella* is characterized by rapid growth if cultured under ideal conditions. On the other hand, *Campylobacter*-contaminated samples showed a slower growth rate, with no exponential growth observed in the considered enrichment time (Figure 1c,d); this is explained by the longer time normally required for this pathogen’s growth.

### 3.2. Colorimetric LAMP Specificity and Inclusivity Tests

The most common hurdle in LAMP assays is unspecific amplification. To verify primer specificity, the colorimetric LAMP for *Salmonella* and *Campylobacter* was tested against other common foodborne pathogens, such as *Y. enterocolitica*, *L. monocytogenes*, and verocytotoxin-producing *E. coli*. As shown in (Figure 2), primers appeared to be specific for the two target pathogens, as DNA amplification (with a subsequent color shift from red to yellow) occurred only in the tubes containing *Salmonella* and *Campylobacter* DNA.

Morover, to verify inclusivity, the primers were tested separately on all strains used for the contamination and on a number of field strains isolated from poultry meat (Appendix A).

### 3.3. Real-Time PCR and LAMP for Salmonella

Our data (Figure 3) highlighted that the real-time PCR could detect *Salmonella* at the lowest concentration tested (10^1^ CFU/g) after shorter enrichment times than those described in the ISO 6579-2:2017 procedure (18 h). Even at the lowest concentrations (10^1^ CFU/g) and at amplification times as short as 45′, 89 out of the 96 positive samples were correctly detected (92.7%, results shown in Appendix A). All amplified samples showed a Ct between 39 and 22 and were therefore considered positive according to AFNOR BRD 07/06–07/04 (Ct ≥ 10). Enrichment incubation times against Ct values were fitted by linear regression for 10^3^ CFU/g and 10^5^ CFU/g concentrations. The Ct values for both concentrations tested were characterized by a significant linear trend (*p* < 0.05), suggesting that replication of the bacteria already takes place in the first hours of enrichment.

Moreover, Ct comparison between the samples and positive control (broth + Sal-monella 105 CFU/g) suggested a low matrix inhibition effect on RT-PCR, since the Ct values for the positive control (105 CFU/g in broth) and for 105 CFU/g contaminated samples were comparable. Regarding the colorimetric LAMP assay, after a 30′ reac-tion, a total of 78 out of 96 samples were found positive (81.3%, see Appendix A). Con-cerning 30′ LAMP, 101 CFU/g contaminated samples were detected at T6 in batch 1 and at T4 in batch 2 (Appendix A), while the 45′ LAMP assay detected in 93 out of 96 samples (96.6%), and the reaction was positive even for the 101 CFU/g T0 samples in batch 1 (Figure 4a) and 101 CFU/g T2 samples for batch 2 (Figure 4b). The real-time PCR was able to detect the same samples, even without any enrichment phase (101 CFU/g T0 (Figure 3a), blue dots). Real-time PCR and colorimetric LAMP results were then compared using McNemar’s and binomial tests. A significant difference (*p* < 0.05) was observed between real-time PCR and 30′ LAMP and between 45′ LAMP and 30′ LAMP in favor of real-time PCR and 45′ LAMP, respectively, while no significant dif-ference was observed between real-time PCR and 45′ LAMP results (*p* > 0.05).

### 3.4. RT-PCR and LAMP for Campylobacter

RT-PCR detected *Campylobacter* starting from a concentration of 10^3^ CFU/g in both minced chicken meat batches, with no need for an enrichment phase (Figure 5, red squares). However, it failed to detect 10^1^ CFU/g contaminated samples at all enrichment times tested (Figure 5, blue dots). *Campylobacter*’s Ct values were barely influenced by the enrichment time (Figure 5) and mirrored the results from pathogen plate count (Figure 1b) according to the pathogen slow growth rate. This was confirmed by the statistical analysis, where no linear trend was observed (*p* > 0.05).

Overall, a total of 71 out of 96 positive samples were correctly detected (74% of the total samples, see Appendix A for summarized results).

As for *Salmonella,* Ct comparison between the samples and positive control (broth + *Campylobacter* 10^5^ CFU/g) suggested a minimal matrix inhibition effect. The 30′ LAMP reaction for *Campylobacter* spp. detection provided results identical to the RT-PCR, with positive outcomes starting from 10^3^ CFU/g at T0 (Appendix A). Negative results were obtained for all 10^1^ CFU/g contaminations (Appendix A). By extending the LAMP reaction for 15 additional minutes, the number of samples detected positive increased (82 out of 96 samples, 82.5% positivity, Appendix A), with 10^1^ CFU/g samples being detected at T4 in batch 1 (Figure 6a). Again, 45′ LAMP and RT-PCR results were compared using McNemar’s and binomial tests, and, in this case, they were significantly different (*p* < 0.05) in favor of the LAMP assay.

## 4. Discussion

To date, most of the time required to detect food pathogens is spent in the enrichment of the microorganism of interest (*Salmonella enterica* and *Campylobacter* spp. in this specific case). Therefore, in our study, we evaluated the reduction of the enrichment phase to shorten the total analysis time without affecting pathogen detection. In particular, the detection of *Salmonella* spp. and *Campylobacter* spp. was assessed after 0, 2, 4, or 6 h, instead of 18 h and 24 h, respectively. A second time-saving step is represented by the implementation of a DNA amplification method that can give results at a glance, without the need for further sample processing (i.e., agarose gel electrophoresis). Contaminated samples were thus analyzed with a routinely used RT-PCR commercial kit and with a colorimetric LAMP. Concerning *Salmonella* detection, both methods proved feasible and had comparable performances, despite 30′ LAMP proving to be slightly lower performing than RT-PCR (*p* < 0.05), since 10^1^ CFU/g of *Salmonella* were detected after 6 h and 4 h of enrichment in batch 1 and 2, respectively (Appendix A) in accordance with plate count results, showing an exponential growth already beginning after 2 h of enrichment (Figure 1a,b). However, 45′ LAMP detected 96.6% positive samples, showing performances comparable to RT-PCR (*p* < 0.05), according to most studies presenting RT-PCR with limits of detection rather similar to LAMP [8]. Moreover, the 45′ assay was positive for the 10^1^ CFU/g samples after a 2 h enrichment, and even without enrichment in batch 1 (Figure 4a,b). These results were consistent with a recent study, which detected after 2 h of enrichment all of the spiked samples as 100% positive by conventional 45′ LAMP [25], while, without the enrichment step, the reported sensitivity varied from 2.2 CFU/g to 10^8^ CFU/mL [8]. However, compared to conventional LAMP, the colorimetric assay enables an easy detection of positive samples without any additional processing or specialist interposition, allowing field-based diagnostics.

Regarding *Campylobacter,* RT-PCR and 30′ LAMP detected *Campylobacter* spp. starting from a concentration of 10^3^ CFU/g without enrichment (Figure 5 and Figure 6). Performing 45′ LAMP significantly increased the number of detected positive samples (from 74% to 82.5%, Appendix A), with 10^1^ CFU/g samples being detected after 4 h of enrichment (Figure 6a). In a study on artificially contaminated swabs, the limit of detection of 60′ LAMP was reported between 10^3^ and 10^4^ CFU/swab [24]. In fact, with our protocol, the increase to 60′ of the colorimetric LAMP resulted in the production of false positives, with a decrease in specificity (data not shown); however, the 30′ or 45′ protocol detected down to 10^3^ CFU/g without enrichment. The results of *Campylobacter* detection with our protocol is consistent with the slow growth rate featured by *Campylobacter* spp., as also demonstrated by plate count results (Figure 1c,d). Indeed, the enrichment phase for *Campylobacter* spp. usually takes more than double the time compared to *Salmonella* (24 h vs. 18 h). Our results suggest that LAMP can outperform real-time PCR in detecting *Campylobacter*, giving a better outcome in half of the time (45′ vs. 90′, *p* < 0.05), and stressing the impact of fast results for the poultry industry, where timely detection is often crucial [20].

With LAMP, the amplification of 10^1^ CFU/g contaminated samples was not always successful, suggesting that this bacterial concentration may be below the limit of detection of this protocol. Overall, our results show that colorimetric LAMP might be suitable for *Salmonella* and *Campylobacter* spp. detection, since, for *Salmonella*, even the lowest concentration is successfully detected after 6 h of enrichment, and, for *Campylobacter*, it is possible to detect 10^3^ CFU/g, meeting the limits set by the EC Regulation No. 2073/2005 regarding *Campylobacter* detection in broiler carcasses [26], potentially providing food business operators a tool for the rapid screening of these pathogens. The present study highlights the possibility to perform a *Salmonella* and *Campylobacter* screening in less than 8 h from specimen receipt, using either RT-PCR or colorimetric LAMP and with minimal differences in terms of sensitivity. Indeed, LAMP is highly specific due to six primers that efficiently amplify the target DNA, resulting in 10^9^ copies in less than 1 h, while RT-PCR produces a DNA amount almost 20 times lesser in about 1–2 h [8,27,28]. However, even if both techniques proved equally good for one-day detection of the foodborne pathogens analyzed, the colorimetric LAMP has the additional advantages of speed, simplicity, and portability. Indeed, four aspects tip the scale in favor of the LAMP technique: (i) the time needed for colorimetric LAMP to obtain a result is half the time needed for RT-PCR (45′ vs. 90′); (ii) the reaction does not require complex instrumentation, since a common heated plate is sufficient (no thermal cycler); (iii) the colorimetric LAMP outcome can be instantly determined by observing the color of the amplification mix (red if negative, yellow if positive, and no fluorescence detectors are needed); and (iv) the risk of environmental contamination by amplified DNA is greatly reduced (the reaction tube does not need to be opened during the detection phase). The second and third points, in particular, give colorimetric LAMP a remarkable edge over RT-PCR, especially if pathogen screening is conducted in environments lacking the proper specialized instrumentation (e.g., for supply chain tests or on-field analyses). Moreover, the naked eye visualization of LAMP products is a promising system already used as a fast and direct diagnostic assay [20,29]. Nevertheless, colorimetric LAMP is not free from uncertainty, as its colorimetric nature under particular conditions (e.g., a very low starting amount of target DNA) leaves space for subjective interpretation when the color shift occurs only partially, featuring orange nuances (see Appendix A). To mitigate this limit, three precautions could be applied: (i) analyze replicates of the same sample and consider it as positive when at least one of them is unquestionably yellow; (ii) increase the starting amount of DNA in the mix; and (iii) further extend the amplification phase by 5 or 10 min.

Regardless of the DNA amplification technique used, the use of a shortened enrichment phase or direct amplification could prove useful in specific situations, such as for self-monitoring purposes or for goods being held for import inspection purposes. Finally, it is worth specifying that this study is a proof of concept, since it was conducted under ideal conditions using experimentally contaminated specimens, but, due to LAMP’s high tolerance to potential assay inhibitors, it is a suitable and robust method to detect pathogens in food matrices at the POC [8]. Further analyses are required to ensure that the methodologies tested here are also fully applicable to matrices other than minced chicken meat and to naturally contaminated samples.

## Figures and Tables

**Figure 1 foods-10-01132-f001:**
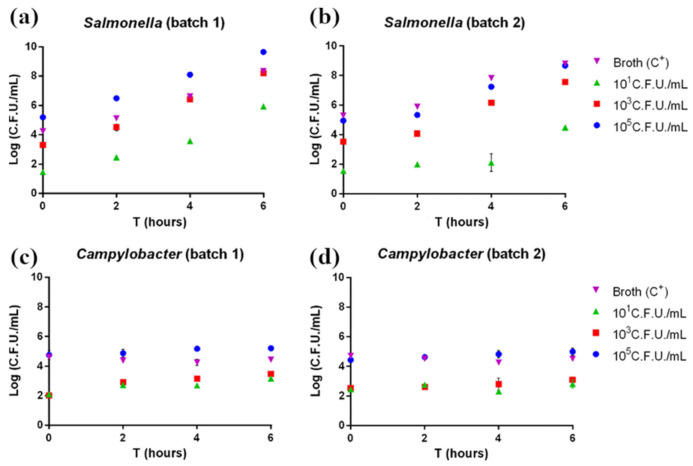
(**a**,**b**) *Salmonella* and (**c**,**d**) *Campylobacter* plate count. Each sample was measured in triplicate. Mean ± SD is shown. The full dataset can be found in Appendix A.

**Figure 2 foods-10-01132-f002:**
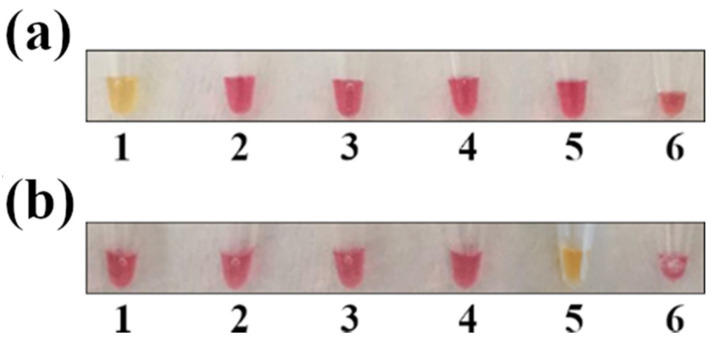
LAMP specificity test. The sample positivity was verified by observing the change in the color of the mix (from red to yellow). (**a**) LAMP for *Salmonella* spp. detection; (**b**) LAMP for *Campylobacter* spp. detection. Legend: 1 = *S. enterica;* 2 = *L. monocytogenes*; 3 = *Y. enterocolitica;* 4 = verocytotoxin-producing *E. coli*; 5 = *C. jejuni;* and 6 = negative control.

**Figure 3 foods-10-01132-f003:**
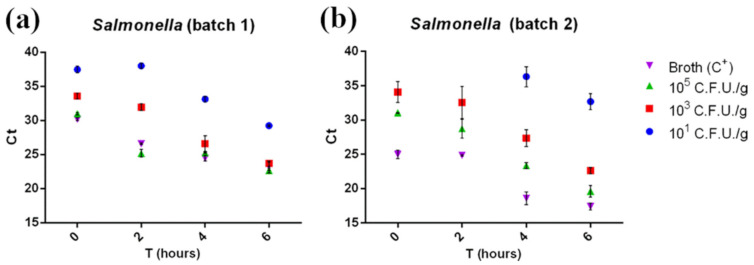
Ct of the first batch (**a**) and the second batch (**b**) of minced chicken meat samples contaminated with *Salmonella* plotted against related enrichment times (0, 2, 4, and 6 h). Ct was inversely correlated with enrichment time and bacterial concentration. No amplification was detected at the 10^1^ CFU/g contamination level T0 and T2 samples in batch 2. Measurements were performed on two separate batches of minced chicken meat in triplicate. The mean ± SD of each measurement is shown. The full dataset can be found in Appendix A.

**Figure 4 foods-10-01132-f004:**
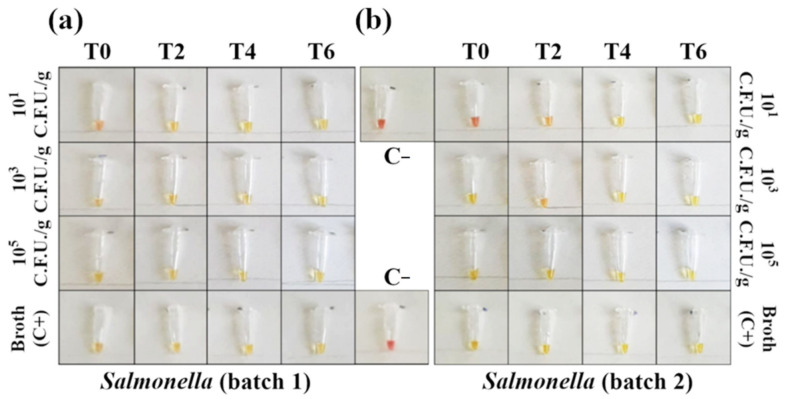
Detection of *Salmonella* DNA in contaminated minced chicken meat using colorimetric LAMP (amplification time 45′). Negative samples are red and positive samples are yellow. (**a**) In the first batch of minced meat, the reaction detected 10^1^ CFU/g with no need for an enrichment phase. (**b**) For the second batch, the reaction amplified 10^1^ CFU/g after 2 h of enrichment. Each sample was tested in triplicate (see Appendix A for the complete panels).

**Figure 5 foods-10-01132-f005:**
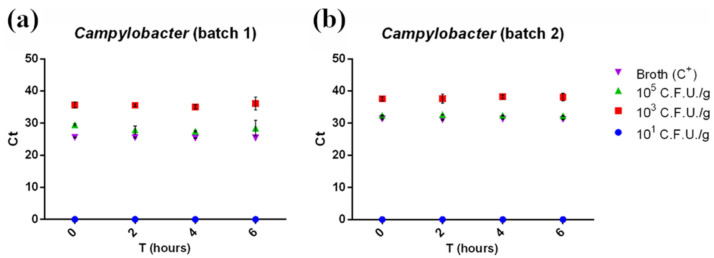
Ct of the first batch (**a**) and the second batch (**b**) of minced chicken meat contaminated with *Campylobacter* spp. plotted against enrichment times (0, 2, 4, and 6 h). Ct was inversely correlated with the starting contamination level, but not with enrichment time, highlighting the slow-growing rate of the pathogen. No amplification was detected in 10^1^ CFU/g contaminated samples. Measurements were performed on two separate batches of minced chicken meat in triplicate. The mean ± SD of each measurement is shown. The full dataset can be found in Appendix A.

**Figure 6 foods-10-01132-f006:**
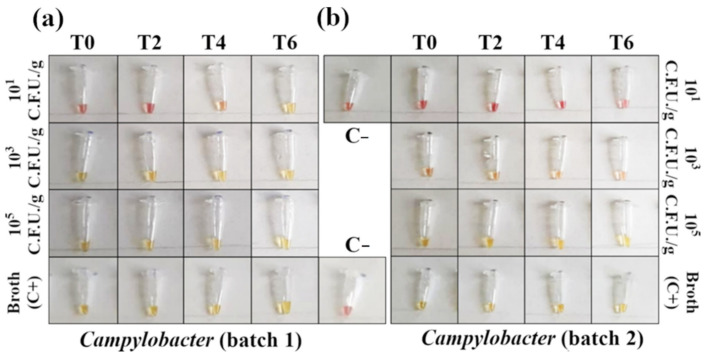
Detection of *Campylobacter* spp. DNA in contaminated minced chicken meat using colorimetric LAMP (amplification time 45′). Negative samples are red and positive samples are yellow. (**a**) In the first batch of minced meat, the reaction detected 10^1^ CFU/g after 4 h of enrichment. (**b**) For the second batch, 10^3^ CFU/g were detected with no need for enrichment. Only one out of three replicates is shown here for each sample (see Appendix A for the complete panels).

**Table 1 foods-10-01132-t001:** List of the ATCC/NCTC *Salmonella* and *Campylobacter* strains used to contaminate the minced chicken meat batches.

	*Salmonella* spp.	*Campylobacter* spp.
Strain 1	*S.* Typhimurium (ATCC 6994)	*C. jejuni* (ATCC 33291)
Strain 2	*S.* Enteritidis (ATCC 13076)	*C. jejuni* (ATCC 49943)
Strain 3	*S*. Infantis (NCTC 6703)	*C. coli* (ATCC 43478)

**Table 2 foods-10-01132-t002:** List of LAMP primers used for the amplification of *Salmonella enterica* and *Campylobacter* spp., and the composition of the 10X primer mix used in the LAMP assays described.

	10X Mix Concentration	Primers for *Salmonella* spp.
FIP	16 µM	3′-TGCACTTTACCGGTACGCTGAATACAGCGGCAATTTCAACCA-5′
BIP	16 µM	3′-CGGTCTGGATTCGCAGGTCAAAGCGATAGCCTGGGGAAC-5′
F3	2 µM	3′-CCGGACAAACGATTCTGGTA-5′
B3	2 µM	3′-CCGACATCGGCATTATCCG-5′
LF	4 µM	3′-TACCCCCTCCGGCTTTTG-5′
LB	4 µM	3′-ACAATGCGTCTTATCGCTACG-5′
		**Primers for *Campylobacter* spp.**
FIP	16 µM	3′-GGACCGTGTCTCAGTTCCAGTGTGACGGATGAGACTATATAGTATCAGCTAG-5′
BIP	16 µM	3′-CGGGAGGCAGCAGTAGGGAATATTGCTAAGAAAAGGAGTTTACGCTCCG-5′
F3	2 µM	3′-CTGCTTAACACAAGTTGAGTAGG-5′
B3	2 µM	3′-TTCCTTAGGTACCGTCAGAA-5′
LF	4 µM	3′-GTTAAGCGTCATAGCCTTGGTAA-5′
LB	4 µM	3′-GCGTGGAGGATGACACTT-5′

## Data Availability

The data in the presented study are available within the article and in the Appendix A.

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
