# Peer review of "One-Day Molecular Detection of Salmonella and Campylobacter in Chicken Meat: A Pilot Study"

_foods, 2021, doi:10.3390/foods10051132_

Round 1
Reviewer 1 Report
The revised version of the manuscript has been improved substantially after the careful revision by the authors.
Author Response
We thank the reviewer for his time and contribution.
Reviewer 2 Report
I would like to thank the authors for the corrections; the manuscript has been significantly improved. However, there are stills some points that need to be clarified.
I’m not sure to understand the authors response regarding the use of Preston or Bolton Broth. In the response to reviewer, authors mentioned : “ we have used the Preston broth for food (poultry meat), this broth is used when in the sample is present competitive indigenous bacteria; while in the same ISO methods the Bolton broth is suggest when in the sample the competitive population is not dominant, and for this reason we have used Bolton to prepare bacterial suspension.” but in the manuscript I understand that bacterial suspensions were prepared in Preston broth, and Bolton broth was used for the contaminated chicken meat.If Preston broth was used for the poultry meat, why 40 h enrichment was used as for this broth an enrichment step of 24 h is recommended for Campylobacter spp?
Minor corrections:
- Paragraph 2.1: Please, write “buffered peptone water (BPW)” the first time you use this abbreviation.
- Paragraph 2.1: Please write 225mL of Bolton broth” instead of 225 ml for homogenization
- Paragraph 3.3: “ A significant difference was observed” Please could explained in favour of which technique as it is specified in paragraph 3.4
Author Response
I would like to thank the authors for the corrections; the manuscript has been significantly improved. However, there are stills some points that need to be clarified.
I’m not sure to understand the authors response regarding the use of Preston or Bolton Broth. In the response to reviewer, authors mentioned : “ we have used the Preston broth for food (poultry meat), this broth is used when in the sample is present competitive indigenous bacteria; while in the same ISO methods the Bolton broth is suggest when in the sample the competitive population is not dominant, and for this reason we have used Bolton to prepare bacterial suspension.” but in the manuscript I understand that bacterial suspensions were prepared in Preston broth, and Bolton broth was used for the contaminated chicken meat.If Preston broth was used for the poultry meat, why 40 h enrichment was used as for this broth an enrichment step of 24 h is recommended for Campylobacter spp?
We apologize with the reviewer for this overlooking, the names of the broths have been swapped during the writing of the manuscript. As such, the meat was enriched in Presto for at most 6 hours. The reference to 40 hours enrichment was corrected to 24 throughout the text.
Minor corrections:
- Paragraph 2.1: Please, write “buffered peptone water (BPW)” the first time you use this abbreviation.
As suggested, the extended name was added to the manuscript.
- Paragraph 2.1: Please write 225mL of Bolton broth” instead of 225 ml for homogenization
As suggested, the unit was corrected.
- Paragraph 3.3: “ A significant difference was observed” Please could explained in favour of which technique as it is specified in paragraph 3.4
As suggested, it has been specified in the text which techniques have the best performances.
Reviewer 3 Report
The authors made the changes suggested by the reviewers.
Author Response
We thank the reviewer for his time and contribution.
This manuscript is a resubmission of an earlier submission. The following is a list of the peer review reports and author responses from that submission.
Round 1
Reviewer 1 Report
Comments:
This manuscript “One-day molecular detection of Salmonella and Campylobacter in chicken meat: a pilot study” evaluated short enrichments combined to a colorimetric Loop-mediated isothermal AMPlification (LAMP) or real-time PCR to detect Salmonella and Campylobacter in poultry meat contaminated at different concentration levels.
Recently some articles have been written about similar subjects:
- Domesle, K. J., Young, S. R., Yang, Q., Ge, B. Loop-Mediated Isothermal Amplification for Screening Salmonella in Animal Food and Confirming Salmonella from Culture Isolation. J. Vis. Exp. (159), e61239, doi:10.3791/61239 (2020).
- Quyen Than Linh, Nordentoft Steen, Vinayaka Aaydha Chidambara, Ngo Tien Anh, Engelsmenn Pia, Sun Yi, Madsen Mogens, Bang Dang Duong, Wolff Anders. A Sensitive, Specific and Simple Loop Mediated Isothermal Amplification Method for Rapid Detection of Campylobacter spp. in Broiler Production. Frontiers in Microbiology. 2019, 10, 2443. 10.3389/fmicb.2019.02443
- Yang Q, Domesle KJ, Ge B. Loop-Mediated Isothermal Amplification for Salmonella Detection in Food and Feed: Current Applications and Future Directions. Foodborne Pathog Dis. 2018;15(6):309-331. doi:10.1089/fpd.2018.2445
So, the topic is not new but, the article, contributes to positive and interesting new data of the subject in question and the introduction summarizes the previous works. Proper experiments were conducted, and the data obtained is remarkably interesting and relevant.
I only have minor comments regarding some microorganisms names that should be in italic (page 4).
I think the article may be published after a minor revision.
Reviewer 2 Report
The aim of this work was to test the effect of short enrichment times on the detection of Salmonella and Campylobacter allowing the detection of these pathogens on poultry meat in a one-day workflow using RT-PCR or LAMP. This objective is clearly important for the poultry industry because detection of these pathogen requires several days using the ISO-methods.
However,the major issue about this work is the lack of discussion of the results. Other similar studies have been performed during the past years to detect Salmonella and Campylobacter on chicken meat and it’s difficult to understand the originality of this particular work without discussing them (for example : Vichaibun and Kanchanaphum, 2020 or Romero and cook, 2018). Moreover, it is important to explain if the objective of the method was to meet the criteria in terms of limit of detection or enumeration that are imposed by the European process hygiene criterion regarding Salmonella or Campylobacter on chicken meat.
You could find below several major comments that must be discuss/correct :
Materials and methods section
- Could you explain why, three different strains of Salmonella and Campylobacter are used in this work?
- Why did you use Preston broth to prepare bacterial suspension and Bolton broth for the enrichment step? In the ISO method for Campylobacter detection (10272 – part 1, 2017) enrichment of Preston broth is recommended for chicken meat, so could you explain why Bolton broth was used? Moreover, Preston broth requires an enrichment for only 24 h, so better results could have been obtained with the 6 hours used in this work. Maybe this could be discussed.
Results section
-Figure 1: The data presented on figure 1a does not correspond to the legend, when comparing to the data presented in tale S2a.
-Colorimetric specificity tests: what about the detection of other Salmonella serovars or other Campylobacter strains? It is explained that 3 strains of each pathogens were used in the bacterial suspension but did you test them separately to prove that the 2 methods could detect different strains of these pathogens?
As the discrimination of results of the LAMP used the naked eyes, could you explain if the result is always red for negative sample or yellow for positive sample because in figure S5 or S6 it seems, it could be orange. How to interpret these results? What is the limit?
Other comments :
Introduction section
- Please update the data about the number of Campylobacteriosis with the latest EFSA report
-Please homogenize using “serovar” throughout the document
Materials and methods section
- Please write the name of bacterial gender in italics throughout the document
- “Then, broths were then re-incubated”, please correct the sentence by “Then, broths were re-incubated”
-In my opinion, table S1 should be in the materials and methods section
- “negative control was diluted in 9 mL PBW” do you mean “PBS” ?
- “centrifugated at 2000 g”, please write “g” in italics
- In my opinion, tables 1 and 2 should be combined, in only one table as the primer names are the same for Campylobacter and Salmonella. The table could regroup on the same raw, the name of the primer, the sequence for Campylobacter, the sequence for Salmonella and the concentration in the mix.
-Tables S2a and S2b should be combined. Same comment for tables S2c and S2d
Reviewer 3 Report
The article “One-day molecular detection of Salmonella and Campylobacter in chicken meat: a pilot study” by Andrea et al., describes here the application of Loop-mediated isothermal AMPlification (LAMP) in rapid detection of Salmonella and Campylobacter in chicken meat.
I have few minor suggestions:
Introduction:
As authors stated that “Loop-mediated isothermal AMPlification (LAMP) is one of the most widely used isothermal method”- Was this method used previously for detection of any food born pathogens and if so please cite it in the manuscript.
Methods:
“To evaluate the ability of the LAMP assay to detect Salmonella and Campylobacter spp. in poultry meat,…..” use italics font for Salmonella and Campylobacter (check it throughout the manuscript).
Then, broths were then re-incubated in fresh media at the same conditions to ensure that most of the cells were in the same physiological state.- How the authors can state that after dilution of the culture, the cells stayed in same physiological state.
Section 2.4: check the unit.